# Using Dried Crickets as a Nutrients and Bioactive Compounds Source in Crispy Vegetable Chips

**DOI:** 10.3390/foods14101810

**Published:** 2025-05-20

**Authors:** Natcharee Jirukkakul, Areeya Phoolklang

**Affiliations:** Division of Applied Science, Faculty of Interdisciplinary Studies, Khon Kaen University, Nong Khai 43000, Thailand; pareeya@kkumail.com

**Keywords:** cricket powder, crispy chips, antioxidants, protein and nutritional value, consumer acceptance

## Abstract

In general, the acceptance of edible insects by consumers is low. Therefore, the aim of this research was to develop protein supplements from desiccated crickets. The objectives of this research were to study the effects of four different drying methods on the chemical properties of crickets and the effects of cricket powder fortification in vegetable chips on the chemical and physical qualities and consumer acceptance. Through an analysis of the chemical composition of cricket powder dried using hot air, vacuum, microwave, and freeze-drying methods, it was found that freeze-drying resulted in the highest protein content in the cricket powder, followed by vacuum drying, hot air drying, and microwave drying. However, the antioxidant activity, which was analyzed using DPPH, showed no significant differences across the four drying methods (*p* > 0.05). The sensory testing of chips by 30 consumers revealed that the chips with a 10:10 ratio of vegetable powder to cricket powder received the highest satisfaction results in all of the test attributes, ranging from “like” to “like very much”. When studying the chemical composition, hardness, and color of the chips, it was found that increasing the amount of cricket powder resulted in a decrease in lightness and yellowness, while redness and hardness increased. The antioxidant activity and phenolic content of the chips increased with the addition of cricket powder, while the flavonoid and potassium contents decreased as vegetable powder was replaced with cricket powder. In the formula most preferred by consumers, the antioxidant activity, phenolic content, flavonoid content, and potassium content were 60.90%, 6.25 ± 0.46 mg GAE/mg sample, 11.16 ± 0.1 mg QE/mg sample, and 0.66 ± 0.01%, respectively.

## 1. Introduction

In 2050, the world population will reach approximately 9.7 billion [1]. Consequently, the demand for alternative protein sources has grown significantly, driven by the increasing global population, environmental concerns, and the rising popularity of health-conscious diets. Edible insects—particularly crickets—have gained attention as a sustainable and highly nutritious protein source. Crickets are widely farmed for food due to their high protein content, which is comparable to that found in pork, chicken, and mackerel. In addition, crickets contain many essential amino acids for the body. They also offer advantages in terms of reducing greenhouse gas emissions and are considered safe from pesticides, chemicals, heavy metals, and allergens [2]. Crickets are classified as a novel food source in Western countries [3] and are regarded as a sustainable alternative protein source [4]. Novel foods must be authorized by the Commission Implementing Regulation (CIR). Food Business Operators (FBOs) have been authorized to place defined types of insect-based products (IBPs) on the EU market. Of course, crickets are some of those insects [1]. The Food Ingredients Asia event showcased food and beverage trends in 2020, highlighting the growing use of alternative and novel proteins, including cricket protein, which had a market value of approximately THB 6.725 billion and was projected to grow by 6.4%, especially in insect-based meat substitute processing. The global edible insect business was reported to have an economic value of USD 400 million [5], with Thailand being a major exporter of edible insects. Insects are a high-protein food that could replace meat in diets. In Thailand, the cricket farming industry is characterized by a family-owned business model with 3000–7000 tons of cricket production per year [6]. Raw materials must come from hygienic farms that are certified by the Department of Livestock Development. Powdering insects is one method of addressing the negative image of consuming insects and marks the beginning of the creation of new foods using insects as ingredients. Crickets are rich in essential amino acids, vitamins, and minerals, making them an ideal candidate for enhancing the nutritional value of various food products. For instance, 100 g of cricket powder provides up to 50% protein, whereas other meats provide only 30% [5]. Insects could also be incorporated into various foods, such as baked goods [7], crackers [8], cereal bars [3], insect snack balls [9], pancakes [10], seasoning sauces [11], pasta [12], biscuits [13], extruded products [14], frankfurters [15], and bread [16]. There is ongoing research related to crickets regarding, for example, the effect of cricket powder particle size on powder nutrients [17], the effect of supplementation on the nutritional value of crickets [18], the evaluation of commercial processing conditions on the nutritional and technological properties of edible crickets [19], and improving the quality of cricket powder to replace wheat flour [20]. This research demonstrates the potential of crickets as an innovative source of protein for industrial production and as a sustainable source of protein for the future.

At present, consumers are looking for snacks that offer variety but are low in fat, sodium, and cholesterol [8]. Snacks are commonly consumed by both children and adults and are considered high-energy foods but offer little nutritional value, as they are rich in starch, sugar, and fat. Since these energy sources are typically sufficient in regular meals, excessive snacking could lead to overconsumption, causing problems such as obesity and tooth decay. Additionally, many market-available snacks contain sodium at levels exceeding the recommended standards, which could harm children’s health in the future. Therefore, snacks that better suit consumer health needs are being developed, with many products incorporating whole grains, vegetables, and fruits, such as potato crisps [21], banana crisps [22], mango crisps [23], and tomato pomace crisps [24]. Using vegetables that are rich in vitamins and minerals to make snacks helps to reduce starch content while increasing dietary fiber, which improves their digestion. Popular fruit and vegetable snacks include fried and crispy options; however, these are still high in fat, which could negatively impact health if consumed in excess. Therefore, baking instead of frying could help to reduce fat content. Increases in nutritional value and bioactive compounds can be derived from the use of additional ingredients while aiming to maintain consumer sensory acceptance. In addition, drying can be performed using microwave, infrared, ultrasound, or vacuum methods, as well as combinations thereof [25]. Crisps can be made from various vegetables, such as mango, pineapple [26], carrot [27], and ivy gourd leaves [28], which enhance their nutritional value, boost antioxidant capacity, reduce starch, and increase fiber.

Kale (*Brassica oleracea* L. Acephala group) belongs to the Brassicaceae family, as do cabbage, broccoli, and cauliflower. Kale is commonly known as the ‘Queen of Greens’ and is recognized as a superfood due to its high and diverse nutritional value compared with other vegetables. In 100 g of fresh kale, there are 384.9 mg of polyphenols and 112.1 mg of vitamin C, with a Trolox equivalent antioxidant capacity (TEAC) of 1175 µM of Trolox. Ferulic acid is the predominant polyphenol, constituting 62% of total polyphenols (240.44 mg/100 g) [29]. Cooked kale retained 78% of the total amino acid content found in fresh leaves [30]. Thus, using kale powder as a snack ingredient offers a healthy alternative.

Developing methods for drying crickets is essential for studying the nutritional value of high-quality cricket powder products. If cricket powder retains its nutritional quality, it can be processed into various high-quality products, boosting consumer confidence and overcoming the negative image associated with insects. Therefore, this study compared drying methods—including hot air drying, vacuum drying, microwave drying, and freeze-drying—in terms of the maintenance of nutritional value. Previously, sun-drying and grinding were common methods. By improving drying efficiency, crickets can be utilized more effectively. Additionally, using potassium-rich kale as a raw material for producing vegetable chips is a novel approach. This would result in crisps with high potassium content, making them suitable for young children and athletes. One such application is incorporating cricket powder into crispy vegetable chips, a popular snack known for its healthy image but typically lacking in protein content. By drying crickets using various techniques and fortifying vegetable chips with cricket powder, it is possible to create a product that not only meets consumer demand for high-protein snacks but also contributes to sustainability. Moreover, fortifying the crisps with cricket protein increases their protein content, adding value to crickets, which are typically sold in dried or powdered form. This research has potential for product development in the food industry and commercialization.

This research aims to explore the effects of four different drying methods—hot air drying, vacuum drying, microwave drying, and freeze-drying—on nutrients and bioactive compounds and to study the effects of cricket powder fortification in vegetable chips on nutrients, bioactive compounds, and consumer acceptance.

## 2. Materials and Methods

### 2.1. Preparation of Cricket Powder

Crickets were provided by Sivaphon Farm, which is a Good Agricultural Practice (GAP)-certified farm in Khon Kaen province, Thailand, indicating hygiene and safety from dangers and contamination. The adult crickets were 40–45 days old upon processing [6]. The crickets were first washed and evenly distributed on trays, followed by drying using one of the following methods: (1) hot air oven drying at 60 °C for 48 h, (2) vacuum drying at 55 °C for 6 h, (3) microwave drying at 550 W for 6.5 min, or (4) freeze-drying. After drying, the crickets were finely ground using a mechanical grinder and passed through a 200-mesh sieve to obtain cricket powder. The resulting powder was stored at 4 °C until further use in subsequent experiments. Kale powder, soy sauce, sugar, cassava starch, pepper powder, and salt were provided by a supermarket (Nong Khai, Thailand).

### 2.2. Preparation of Cricket Protein-Enhanced Chips

The ingredients were weighed according to Table 1. The ingredients were mixed in a pan, cooked over low heat (70 °C), and stirred constantly for 5 min until the mixture thickened. The mixture was poured into a tray lined with a silicone mat and spread onto a thin sheet. Then, it was dried in a hot air oven at 70 °C for 1 h, removed, and cut into sheets of the desired size. Finally, it was dried for an additional 2 h.

### 2.3. Physical Analysis of Chips

#### 2.3.1. Color Analysis

Color analysis of chips was performed using a Hunter Lab colorimeter (Model TC-P III A, Tokyo Denshoku Co., Ltd., Japan) by measuring each sample 5 times. The values obtained are expressed as L*, a*, and b*: L* (the lightness factor) to indicate brightness. As the L* value of an object approaches zero, it becomes darker. If the L* value is near 100, the object is brighter. a* represents red and green colors. The b* value represents yellow and blue colors.

#### 2.3.2. Hardness

The hardness of the chips was measured using a Texture Analyzer (Model TA.XT plus, Stable Micro Systems, Ltd., Godalming, UK) with a head cutting plate. The machine settings were pre-test, test, and post-test speeds of 1.5, 2, and 10 mm/s, respectively. The hardness of the sample was recorded in Newtons (N), and the hardness of each sample was measured 10 times.

### 2.4. Chemical Analysis of Cricket Powder, Vegetable Powder, and Chips

#### 2.4.1. Composition Analysis

The proximate composition analysis of cricket powder and chips included fat, protein, ash, fiber, and carbohydrates according to the AOAC standards [31], with three repetitions per sample.

#### 2.4.2. Antioxidant Activity of Cricket Powder, Vegetable Powder, and Chips

The antioxidant activity of the cricket powder, vegetable powder, and chips was tested using the DPPH method. The DPPH sample was prepared using Equation (1) to calculate the weight of DPPH in proportion to the volume to be used, adjusting with methanol:(1)$$g=\frac{m1\times v1\times Mw}{1000}$$
where
*g* = weight of DPPH;*m*1 = DPPH concentration;*v*1 = required volume;*Mw* = molecular weight (394.4).

Next, 4 g of sample was dissolved in 3 mL of hexane for 15 h. The clear solution (0.5 mL) was extracted and mixed with 1.5 mL of ethyl acetate. Afterward, 1.5 mL of the solution was mixed with 1.5 mL of DPPH, incubated in the dark for 30 min, and then measured using UV at 515 nm. The antioxidant activity (AA%) was calculated using the following equation:(2)$$AA\%=\frac{\left(Acontrol-Asample\right)\times 100}{Acontrol}$$
where
*AA*% = antioxidant activity;*Acontrol* = control UV reading;*Asample* = sample UV reading.

For the control, 1.5 mL of ethyl acetate was mixed with 1.5 mL of DPPH, incubated in the dark for 30 min, and then measured using UV at 515 nm. The antioxidant was measured in 3 repetitions per sample.

#### 2.4.3. Total Phenolic and Flavonoid Content of Vegetable Powder and Chips

The total phenolic content was analyzed using the Folin–Ciocalteu method. Distilled water (5 mL) was mixed with 0.2 mL of extract and 0.5 mL of Folin–Ciocalteu reagent and left for 5 min in the dark, and then 1.5 mL of Na_2_CO_3_ (75 g/L) was added. The mixture was kept in the dark for 1 h and 30 min. The absorbance was measured at 725 nm, with methanol (99%) used as a blank. The total phenolic content was compared with a gallic acid standard curve [32].

The total flavonoid content was analyzed using the aluminum nitrate method. Distilled water (2 mL) was mixed with 0.3 mL of extract and 0.5 mL of 5% NaNO_2_ and left for 5 min, and then 0.15 mL of 10% Al(NO_3_)_3_ was added and left for another 5 min, followed by 0.15 mL of 1 M NaOH. The absorbance was measured at 420 nm, with methanol (99%) as a blank. The total flavonoid content was compared with a quercetin standard curve [33]. The total phenolic and flavonoid contents were measured in 3 repetitions per sample.

#### 2.4.4. Potassium Analysis of Kale Powder and Chips

To analyze the potassium content, 0.5 g of vegetable powder or chips was weighed into a 100 mL beaker. Then, 5 mL of 50% *v*/*v* HNO_3_ was added and digested at 95 °C for 15 min. After cooling, 10 mL of concentrated HNO_3_ was added and digested again at 95 °C for 1 h or until fully dissolved. Once cooled, the solution was transferred to a 100 mL volumetric flask, diluted with distilled water, and filtered through No. 1 filter paper. The filtered solution was analyzed for potassium using atomic absorption spectrophotometry (AAS) (Model PinAAcle 900F, PerkinElmer Inc., Springfield, IL, USA).

### 2.5. Sensory Evaluation of Chips

This research was approved for human research ethics from the Khon Kaen University Ethics Committee for Human Research (KKUEC) in KKUEC’s Exemption Determination Regulations No. HE68018. General consumers, consisting of 30 students from Khon Kaen University, Nong Khai Campus, were used to study consumer acceptance of 5 formulations of chips. The evaluation was conducted using a 9-point Hedonic Scale, where 1 represents “dislike extremely” and 9 represents “like extremely”. The characteristics assessed included appearance, color, odor, flavor, and overall liking. The chips were packaged in sealed plastic bags and evaluated in a laboratory under white light.

### 2.6. Preservation Testing of Chips

The chips that received the highest sensory acceptance were wrapped in rice berry film mixed with marigold extract [34] and stored for shelf-life testing. TBARSs (Thiobarbituric Acid Reactive Substances) were measured every month for 6 months to assess lipid oxidation. The analysis of lipid oxidation (TBARSs) was conducted by weighing 2 g of the sample and placing it in a test tube (the blank was prepared using an empty tube without the sample). Then, 3 mL of TBA solution was added, and the mixture was vortexed. Next, 17 mL of TCA solution was added, and the sample was boiled for 30 min. After cooling to room temperature, 5 mL of the clear supernatant was extracted and placed in a 50 mL centrifuge tube. Then, 5 mL of chloroform was added, and the mixture was vortexed in a fume hood. The sample was centrifuged at 200× *g* for 5 min, and 3 mL of the supernatant was extracted into a 15 mL centrifuge tube. Next, 3 mL of petroleum ether was added, and the mixture was vortexed in a fume hood. The sample was centrifuged at 200× *g* for 10 min, and the lower layer was extracted using a dropper and measured for absorbance at 532 nm using a spectrophotometer, with distilled water as the blank (set to zero).

The moisture content of the samples was determined using a Moisture Balance (Mettler-Toledo GmbH, model HE73, Greifensee, Switzerland). Aluminum trays were preheated at 105 °C and subsequently cooled before being loaded with the samples. The instrument automatically calculated the moisture content based on weight loss during heating.

Color evaluation was performed using a Hunter Lab colorimeter (Model TC-P III A, Tokyo Denshoku Co., Ltd., Tokyo, Japan). Samples were placed into test cups and positioned at the measurement port of the instrument. Color parameters were recorded according to the CIE Lab color system.

For microbiological determination, a homogenized food sample (1 mL) was aseptically pipetted into 9 mL of sterile distilled water to obtain a 10^1^ dilution. Serial dilutions were prepared up to 10^3^ following aseptic techniques. From each dilution, 0.1 mL was pipetted onto Plate Count Agar (PCA) and spread evenly across the surface using an aseptic technique. Each dilution level was tested in triplicate (3 plates per dilution). Plates were incubated at 35–37 °C for 24–48 h. Following incubation, the total number of microbial colonies was counted, and the microbial load of the original sample was calculated.

### 2.7. Statistical Analysis

The color analysis and hardness were measured in 5 and 10 repetitions, respectively. Meanwhile, the chemical analysis and preservation testing were performed in 3 repetitions per sample. The results from both parts of the experiment—the first being the 4 drying methods for crickets and the second being the 5 formulations of chips—were analyzed using a randomized complete factorial design. Data variance was analyzed using a one-way ANOVA test, and the differences between the 4 conditions of cricket powder and the 5 formulations of chips were compared using Duncan’s Multiple Range Test at a confidence level of *p* < 0.05 with the SPSS software (version 23).

## 3. Results and Discussion

### 3.1. Physical and Chemical Analysis of Cricket Powder

From the analysis of the chemical composition of cricket powder dried using hot air, vacuum, microwave, and freeze-drying methods, it was found that the microwave drying method resulted in the lowest moisture and protein content, while freeze-drying led to the highest protein content but the lowest carbohydrate content. This is because freeze-drying is a non-thermal drying method that preserves nutritional value, including that pertaining to proteins, which are not denatured by heat. In contrast, the high heat from microwave drying resulted in the highest temperature in the final dried crickets, which was 80–90 °C, resulting in the lowest protein content in the cricket powder. Vacuum drying, which had a lower drying temperature than atmospheric conditions, produced higher protein content in the cricket powder compared with hot air drying. The total protein content ranged from 51.44 to 58.51%, which was consistent with other research, where the protein content was between 48.06 and 76.19% [35], but lower than the studies of Machado and Thys [36] and Cheng et al. [37], who reported 62.76% and 66%, respectively. Nevertheless, the protein content in the cricket powder was higher than that in beef (25.6%), milk powder (26.3%), and chicken (39%). The nitrogen-to-protein conversion factor of insect species is 5.33 because the accurate true protein content is calculated from nitrogen after the deduction of nitrogen in chitin. However, the protein content in crickets remains high and contains large amounts of essential amino acids [6] (Table 2).

The fat content of the cricket powder ranged from 26.85 to 28.98%, with the microwave drying method resulting in the highest fat content because more water evaporated than in other methods due to higher temperatures, while the other methods did not show significant statistical differences (*p* > 0.05), and there was an inverse relationship with moisture content. Typically, the fat content of crickets ranged between 20.4 and 29.3%, with the lipid content consisting of sterols, waxes, monoglycerides, diglycerides, triglycerides, and phospholipids. The vitamins found included A, D, E, and K. The fatty acids present were oleic, linoleic, stearic, and palmitic acid [6]. The oxidation of cricket powder could be reduced by ginger and garlic extract [38].

The fiber content of the cricket powder ranged from 10.34 to 12.73%, which was higher than that of the cricket powder of Machado and Thys [36] (3.70–7.50%). This also depended on the species and age of the crickets. Lower drying temperatures resulted in reduced fiber content in the crickets. When crickets were dried using vacuum and freeze-drying methods, the fiber content was lower than when dried with hot air and microwave methods (*p* < 0.05).

The ash content of the cricket powder ranged from 3.04 to 3.40%, which was similar to that in the cricket powder of Machado and Thys [36], which had a value of 3.19%. The ash content indicated the mineral content in the cricket powder, and some minerals could be destroyed by heat. Thus, freeze-drying resulted in the highest ash content, followed by vacuum drying, hot air drying, and microwave drying, respectively.

The carbohydrate content of the cricket powder ranged from 0.12 to 4.09%, which was similar to that in the cricket powder of Pilco-Romero et al. [35] (1.60–10.20%). Carbohydrates in crickets were present in the form of glycogen, which is used during metamorphosis and male stridulation, due to metabolic interconversion. Crickets have a high carbohydrate content compared with other animal tissues [6].

The antioxidant activity analyzed using DPPH ranged from 76.04 to 76.74%, with no significant differences among the four drying methods (*p* > 0.05), as shown in Table 3. This was because heat does not affect antioxidant activity. Similarly, in an experiment on drying crickets at 140–200 °C for 10 min, it was found that the DPPH values did not differ [4]. However, the antioxidant activity increased when switching from water extraction to ultrasound-assisted extraction (UAE) with 100% or aqueous ethanol, where the DPPH values reached 80–90%, which was higher than with water extraction. Additionally, the antioxidant activity could increase depending on the food that crickets consume.

### 3.2. Sensory Evaluation of Chips

A sensory evaluation of chips by 30 panelists demonstrated that the overall preference for all samples was not significantly different (*p* > 0.05), but the chips without cricket powder were most accepted in terms of appearance and color, as they had the brightest green color. However, there was no significant difference from the samples with 15:5 and 10:10 ratios of vegetable powder to cricket powder (Table 4). On the other hand, the chip with a kale–cricket powder ratio of 0:20 received the lowest satisfaction scores for appearance and color, with statistical significance (*p* < 0.05), as the cricket powder resulted in a brown color (Figure 1). This result was consistent with the color measurement findings, where the lightness (L*) and yellowness (b*) values decreased while the redness (a*) increased as the amount of cricket powder increased. Cricket powder chips ranged from neutral to slightly liked in all sensory attributes. Meanwhile, the chips containing a 10:10 ratio of vegetable powder to cricket powder or 20% each of vegetable and cricket powder among the total dry ingredients were the most liked in all sensory attributes, ranging from liked to very liked. Specifically, the flavor of the kale–cricket chip with a 10:10 ratio received significantly higher scores (*p* < 0.05) than the samples with 20:0 and 15:5 ratios. As for texture satisfaction, it varied by initially increasing and then decreasing with higher amounts of cricket powder, with the 10:10 ratio chip achieving the highest score.

Thus, the vegetable chips with a 10:10 ratio of vegetable powder to cricket powder were selected for further study, focusing on chemical composition and color comparison with vegetable chips (that did not contain cricket powder) and cricket powder chips (that did not contain kale powder).

### 3.3. Chemical Analysis of Vegetable Powder and Chips

Moisture, protein, fat, ash, and fiber are key chemical components for food products. Moisture is an indicator of product quality and shelf life since high moisture levels can trigger chemical reactions such as oxidation and promote microbial growth, in addition to causing cause physical changes. All samples had moisture contents ranging from 0.95 to 1.03%, which was very low (below 13%), classifying them as dry foods that are safe from microbial spoilage [39] (Table 5). However, they should still be stored in packaging that limits air permeability to prevent the absorption of moisture from the atmosphere, which could alter the texture of the food. Additionally, exposure to oxygen can lead to rancidity, and as moisture increases, so does the potential for microbial growth. Most snacks, such as mango snacks (7%) [23], carrot snacks (5%) [40], vegetable bars (0.3–1.2%) [41], and vegetable snacks (3–4%), have low moisture content to maintain crispness [42].

The protein content of chips increased as more cricket powder was added, reaching its highest content when vegetable powder was entirely replaced by cricket powder (9.13–16.39%). All samples had protein levels 3–4 times higher than those of typical snacks, such as corn and green bean snacks, which contain 2.9–3.7% and 2.4% protein, respectively [43]. This shows that these chips are a healthy product for those seeking higher protein intake.

The fat content also increased with the amount of cricket powder, as crickets are rich in both protein and fat. However, the fat content remained lower than in other snacks, such as potato crisps, which have 34–46% fat [21]. The fat content decreases as the frying oil temperature increases, as most snacks are deep-fried, resulting in higher fat content. Additionally, it depends on the type of oil used. In this experiment, the chips were made using a baking process instead of frying, which resulted in low-fat chips. This meets the needs of health-conscious consumers and helps address childhood obesity, a current nutritional issue.

The ash content indicates the mineral levels in the samples, with kale being rich in potassium. The chips had high ash content, which decreased when vegetable powder was replaced with cricket powder. This makes these chips a healthy food that is rich in minerals and suitable for children or those needing mineral supplementation.

The fiber content of the chips was approximately 3%, which was higher than that in typical snacks, such as bamboo shoot snacks, which had 0.68–1.21% fiber [44].

The antioxidant activity and phenolic content of the chips increased as more cricket powder was added, since crickets have high antioxidant properties. However, the flavonoid and potassium content decreased when vegetable powder was replaced with cricket powder. In the formulation most preferred by consumers, the antioxidant activity, phenolic content, flavonoid content, and potassium were 60.90%, 6.25 ± 0.46 mg GAE/mg sample, 11.16 ± 0.1 mg QE/mg sample, and 0.66 ± 0.01%, respectively (Table 6).

### 3.4. Physical Analysis of the Chips

The lightness of the chips ranged from 27 to 32, decreasing as more cricket powder was added, giving it a darker hue. The redness value was low, close to 0, as the chips were green and became redder with the addition of cricket powder, which had a reddish-brown color. The yellowness value ranged from 5 to 17. When compared with kale film [29] and papaya film [45], the chips had lower lightness than both but had similar redness to papaya film and similar yellowness to kale film. The low lightness was due to the addition of ingredients such as sauce and sugar, as well as the heating process, which made the chips darker and more opaque. The lightness was higher, but the redness was lower than in red cabbage crisps because of the ingredient color and the thickness of the crisps [46]. However, the lightness, redness, and yellowness values were higher than those of ivy gourd chips [28], which had lightness, redness, and yellowness values of 25.63, −5.61, and 7.54, respectively.

Adding cricket powder to the chips significantly affected their hardness (*p* < 0.05), with the chips without cricket powder having a hardness of 2.55 N (Table 7), which was higher than that of ivy gourd chips (0.46 N) [28]. This was due to ingredients such as sugar and starch in the chips, which acted as binding agents in the chip structure, increasing their hardness. However, since the chips were thin, their hardness was lower than that of other larger snacks, such as crispy banana [22] and potato crisps [21], which had hardness values of 25–35 N and 14–23 N, respectively.

### 3.5. Preservation Testing of the Chips

The chips that received the highest sensory satisfaction, which had a 10:10 ratio of vegetable powder to cricket powder, were wrapped in rice berry film mixed with marigold extract and stored. TBARSs (Thiobarbituric Acid Reactive Substances) were checked every month for 6 months. It was found that oxidation increased significantly during the first 4 months, representing 40% of the initial value. After 6 months of storage, the oxidation had increased by no more than 50% (Table 8).

Upon checking the color values, the lightness, redness, and yellowness decreased significantly from the initial day (*p* < 0.05). However, consumers were unable to distinguish these changes based on appearance alone.

The moisture content and total microbial counts were examined. It was found that the moisture content was very low (1%), with no significant difference from the product before storage, which corresponded with the absence of microbial growth.

Regarding microbial growth analysis, the total microbial count was no more than 4 log CFU/g, and no yeast or mold was detected. This meets the standard for crispy seaweed products [47], which was the closest comparable product to the one in this experiment. Therefore, the shelf life of the chips should not exceed 6 months.

However, to reduce color changes and oxidation during the storage of chips, it is recommended to use vacuum packaging, opaque containers, and antioxidants. These measures could extend the shelf life beyond 6 months.

## 4. Conclusions

Through a chemical composition analysis of cricket powder, it was found that freeze-drying resulted in the highest protein content. However, the antioxidant activity, which was analyzed through DPPH, showed no significant differences across the four drying methods (*p* > 0.05). For commercial applications, freeze-drying is recommended for the preparation of cricket powder. However, for small-scale industrial processing, other drying methods may be used to help reduce costs.

When the cricket powder was applied to chips, it was found that increasing the amount of cricket powder decreased lightness and yellowness while increasing redness and hardness. The antioxidant activity and phenolic content of the chips increased with the amount of cricket powder added, as crickets have high antioxidant properties. Meanwhile, the flavonoid and potassium contents decreased when vegetable powder was replaced with cricket powder. Notably, the chips containing a 10:10 ratio of vegetable powder to cricket powder received the highest satisfaction ratings in all attributes. This product can be stored safely for 6 months with safe oxidation reactions and no microbial growth.

The formula that achieved the highest consumer satisfaction had high protein and fiber content as well as a lower fat content. It also exhibited high antioxidant activity, making vegetable chips with cricket powder a promising healthy snack alternative and a functional food that meets the demands of future food trends.

## 5. Patents

Natcharee Jirrukkakul and Areeya Phoolkang, 2024. Vegetable sheet crisp formulation. Thai petty patent No. 2403003266.

## Figures and Tables

**Figure 1 foods-14-01810-f001:**
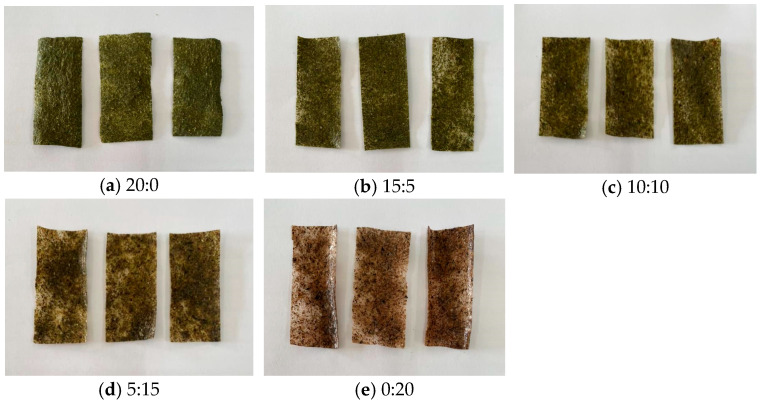
The appearance of the chips (kale powder–cricket powder): (**a**) 20:0, (**b**) 15:5, (**c**) 10:10, (**d**) 5:15, and (**e**) 0:20.

**Table 1 foods-14-01810-t001:** Chip formulations.

Ingredients (g)	Kale Powder/Cricket Powder				
	20:0	15:5	10:10	5:15	0:20
Kale powder	20	15	10	5	0
Cricket powder	0	5	10	15	20
Water	200	200	200	200	200
Soy sauce	1	1	1	1	1
Sugar	26	26	26	26	26
Cassava starch	20	20	20	20	20
Pepper powder	0.5	0.5	0.5	0.5	0.5
Salt	1	1	1	1	1

**Table 2 foods-14-01810-t002:** Chemical composition of the cricket powders created using the four drying methods.

Composition	Hot Air Cricket Powder	Vacuum-Dried Cricket Powder	Microwaved Cricket Powder	Freeze-Dried Cricket Powder
Moisture content (wb)	2.87 ± 0.03 ^b^	3.26 ± 0.52 ^a^	1.42 ± 0.25 ^c^	3.24 ± 0.06 ^a^
Fat (db)	26.94 ± 0.19 ^b^	26.85 ± 0.24 ^b^	28.98 ± 0.32 ^a^	27.13 ± 0.61 ^b^
Protein (db)	53.45 ± 0.02 ^c^	55.80 ± 0.08 ^b^	51.44 ± 0.67 ^d^	58.51 ± 0.12 ^a^
Fiber (db)	12.31 ± 0.24 ^b^	10.34 ± 0.20 ^d^	12.73 ± 0.07 ^a^	10.84 ± 0.11 ^c^
Ash (db)	3.21 ± 0.02 ^c^	3.31 ± 0.02 ^b^	3.04 ± 0.04 ^d^	3.40 ± 0.01 ^a^
Carbohydrate (db)	4.09 ± 0.28 ^a^	3.70 ± 0.39 ^b^	3.81 ± 0.28 ^a^	0.12 ± 0.03 ^c^

Different letters in the vertical line indicate significant differences (*p* < 0.05).

**Table 3 foods-14-01810-t003:** Antioxidant activity of cricket powders created using four drying methods.

Sample	DPPH (%)
Hot air oven cricket powder	76.22 ± 0.37
Vacuum-dried cricket powder	76.56 ± 0.54
Microwaved cricket powder	76.04 ± 0.61
Freeze-dried cricket powder	76.74 ± 0.80

**Table 4 foods-14-01810-t004:** Sensory evaluation of chips (kale powder–cricket powder).

Kale–Cricket	Appearance	Color	Odor	Flavor	Texture	Overall
20:00	7.17 ± 1.21 ^a^	7.17 ± 1.09 ^a^	6.10 ± 1.27 ^a^	5.57 ± 1.52 ^b^	5.50 ± 1.74 ^c^	5.77 ± 1.48 ^a^
15:05	7.00 ± 0.95 ^a^	6.67 ± 0.92 ^ab^	5.90 ± 1.37 ^ab^	5.63 ± 1.59 ^b^	6.33 ± 1.95 ^abc^	5.77 ± 1.72 ^a^
10:10	6.90 ± 0.92 ^a^	7.00 ± 1.10 ^ab^	6.53 ± 1.38 ^a^	6.63 ± 1.71 ^a^	7.07 ± 1.57 ^a^	6.57 ± 1.65 ^a^
5:15	6.90 ± 1.03 ^a^	6.47 ± 1.25 ^b^	6.17 ± 1.15 ^a^	6.33 ± 1.69 ^ab^	6.60 ± 1.59 ^ab^	6.37 ± 1.47 ^a^
0:20	6.03 ± 1.45 ^b^	5.57 ± 1.57 ^c^	5.40 ± 1.28 ^b^	5.50 ± 1.55 ^b^	5.90 ± 1.75 ^bc^	5.67 ± 1.49 ^a^

Different letters in a column indicate significant differences (*p* < 0.05).

**Table 5 foods-14-01810-t005:** Chemical composition of the chips (kale powder–cricket powder).

Kale–Cricket	Moisture Content (wb)	Protein (db)	Fat (db)	Ash (db)	Fiber (db)
20:0	0.95 ± 0.20 ^a^	9.13 ± 0.24 ^c^	0.16 ± 0.12 ^c^	7.22 ± 0.28 ^a^	3.37 ± 0.09 ^a^
10:10	1.01 ± 0.07 ^a^	13.27 ± 0.31 ^b^	1.15 ± 0.21 ^b^	5.14 ± 0.05 ^b^	2.98 ± 0.11 ^b^
0:20	1.03 ± 0.02 ^a^	16.39 ± 0.32 ^a^	2.36 ± 0.12 ^a^	2.70 ± 0.05 ^c^	2.96 ± 0.03 ^b^

Different letters in a column indicate significant differences (*p* < 0.05).

**Table 6 foods-14-01810-t006:** Antioxidant activity and phenolic, flavonoid, and potassium contents of the chips (kale powder–cricket powder).

Kale–Cricket	DPPH (%)	Phenolic Content (mg GAE/mg Sample)	Flavonoid Content (mg QE/mg Sample)	Potassium (%)
20:0	10.09 ± 0.63 ^c^	3.76 ± 0.23 ^c^	12.15 ± 0.26 ^a^	1.66 ± 0.30 ^a^
10:10	60.90 ± 2.23 ^b^	6.25 ± 0.46 ^b^	11.16 ± 0.19 ^b^	0.66 ± 0.01 ^b^
0:20	93.13 ± 0.31 ^a^	8.43 ± 0.32 ^a^	6.88 ± 0.10 ^c^	0 ± 0.00 ^c^
Kale powder	32.62 ± 0.58	8.57 ± 0.63	21.25 ± 0.43	5.34 ± 1.58

Different letters in a column indicate significant differences (*p* < 0.05).

**Table 7 foods-14-01810-t007:** Color and hardness of chips (kale powder–cricket powder).

Kale–Cricket	L*	a*	b*	Hardness (N)
20:0	31.86 ± 1.00 ^a^	0.28 ± 0.11 ^c^	16.48 ± 0.95 ^a^	2.56 ± 0.02 ^c^
10:10	28.64 ± 2.16 ^b^	0.95 ± 10.09 ^b^	13.34 ± 1.19 ^b^	5.28 ± 0.03 ^b^
0:20	27.07 ± 2.30 ^b^	3.33 ± 0.27 ^a^	5.74 ± 0.68 ^c^	14.42 ± 0.27 ^a^

Different letters in a column indicate significant differences (*p* < 0.05).

**Table 8 foods-14-01810-t008:** Color and oxidation of chips with kale powder and cricket powder in a ratio of 10:10.

Storage Time (Month)	L*	a*	b*	TBARS Increase (%)
0	29.00 ± 0.20 ^b^	2.88 ± 0.06 ^a^	14.33 ± 0.22 ^a^	0 ± 0.00 ^e^
1	29.85 ± 1.02 ^b^	1.66 ± 0.22 ^b^	11.67 ± 0.87 ^b^	4.53 ± 1.23 ^d^
2	28.18 ± 1.89 ^b^	1.16 ± 0.15 ^c^	10.96 ± 1.40 ^b^	11.73 ± 2.48 ^c^
3	30.29 ± 0.26 ^b^	0.92 ± 0.01 ^d^	11.56 ± 0.03 ^b^	23.50 ± 0.27 ^b^
4	29.15 ± 0.33 ^b^	0.47 ± 0.02 ^e^	9.89 ± 0.40 ^c^	37.70 ± 11.58 ^a^
5	31.54 ± 0.05 ^a^	0.25 ± 0.01 ^f^	12.31 ± 0.03 ^b^	40.90 ± 2.31 ^a^
6	31.13 ± 0.01 ^a^	0.10 ± 0.01 ^g^	8.75 ± 0.02 ^d^	45.13 ± 5.49 ^a^

Different letters in a column indicate significant differences (*p* < 0.05).

## Data Availability

The original contributions presented in this study are included in the article; further inquiries can be directed to the corresponding author.

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
