# Peer review of "Using Dried Crickets as a Nutrients and Bioactive Compounds Source in Crispy Vegetable Chips"

_foods, 2025, doi:10.3390/foods14101810_

Round 1

Reviewer 1 Report

Comments and Suggestions for Authors

This study aimed to evaluate the effects of different drying methods on the chemical composition of cricket powder and to determine the optimal ratio of cricket to vegetable powder in chips for consumer preference, nutritional value, and antioxidant properties. The study showcases an innovative approach to enhancing the nutritional value of snack foods using cricket powder, however, the methodology and presentation of results need to be improved.

  1. The abstract lacks a clearly stated aim of the work.
  2. Can the Authors justify the use of only the DPPH assay for measuring antioxidant activity, and explain why additional complementary methods (e.g., ABTS, FRAP, ORAC) were not included to strengthen the reliability of the results?
  3. Can the Authors specify how many replicates were performed for the measurements of antioxidant activity, total flavonoid content, and total phenolic content?
  4. The major concern lies in the sensory evaluation process, as it appears that the panel consisted of untrained, randomly selected individuals, which raises questions about the reliability of the sensory data. Additionally, the Authors should clarify whether ethical approval was obtained and if participants provided informed consent, in accordance with bioethical standards for human subject research.

Author Response

1 This study aimed to evaluate the effects of different drying methods on the chemical composition of cricket powder and to determine the optimal ratio of cricket to vegetable powder in chips for consumer preference, nutritional value, and antioxidant properties. The study showcases an innovative approach to enhancing the nutritional value of snack foods using cricket powder, however, the methodology and presentation of results need to be improved.

Response: The methodology and results have been improved.

2 The abstract lacks a clearly stated aim of the work.

Response: The sentences have been added in abstract. “This research was based on the low acceptance of edible insects, so the development of protein supplements from crickets was interesting. The objectives of this research were to study the effects of different drying 4 methods on the chemical properties and were to study the effects of cricket powder fortification in vegetable chips on the chemical and physical qualities and consumer acceptance.”

3 Can the Authors justify the use of only the DPPH assay for measuring antioxidant activity, and explain why additional complementary methods (e.g., ABTS, FRAP, ORAC) were not included to strengthen the reliability of the results?

Response: Since crickets contain fats that are soluble in organic solvents, the DPPH method is suitable. Additionally, it is a fast and straightforward method that does not require pH control. This analysis was conducted to compare samples with different amounts of cricket content.

4 Can the Authors specify how many replicates were performed for the measurements of antioxidant activity, total flavonoid content, and total phenolic content?

Response: Three replicates per sample have been used for antioxidant activity, total flavonoid content, and total phenolic content measurement. The information was added in the manuscript.

5 The major concern lies in the sensory evaluation process, as it appears that the panel consisted of untrained, randomly selected individuals, which raises questions about the reliability of the sensory data. Additionally, the Authors should clarify whether ethical approval was obtained and if participants provided informed consent, in accordance with bioethical standards for human subject research.

Response: The ethics information has been added in sensory evaluation part. “This research has been approved by human research ethics from the Khon Kaen University Ethics Committee for Human Research (KKUEC) in KKUEC’s Exemption Determination Regulations No. HE68018.”

Reviewer 2 Report

Comments and Suggestions for Authors

Dear Authors,
my comments on the manuscript are presented below.

Introduction: I propose to broaden the thematic scope of this chapter. Important from the point of view of developing new products with the addition of cricket powder is a thorough and comprehensive market research. And information on the subject should be included in the introduction. In addition, it is also worth mentioning the obstacles to introducing cricket powder into food production. Is it regulated by law? What are the concerns? e.g. microbiological safety of such a raw material? Any antinutritional components?
Material and Methods
I would like more information about where the crickets came from (what farm, "age" of the crickets).
Hardness measurements were taken. Were crispiness measurements taken of the chips?
How was the product described for sensory testing? Were respondents recruited using a recruitment form to identify a group of respondents with a positive/moderate attitude towards the introduction of cricket powder into food products?
Storage tests (6 months) should also concern sensory characteristics, water content, water activity, and colour and texture.
Results and Discussion: The discussion of the results is correct, however, following the comments on the methods, it is necessary to broaden the issue of storage studies. Such research and its results would contribute to proposing an optimal method of packaging this product. The authors focus only on TBARS and colour measurements. 
Conclusions: The conclusions in the text are a repetition of the discussion of the obtained results. In the meantime, at this point it should be concluded whether the used additive is promising as a component of the recipe for chips? To what extent does it change the nutritional value of the product at the level of the best consumer acceptability? What should be improved so that the share in the recipe is greater and at the same time the product gains consumer acceptance. And very important storage studies, to which the authors referred only to a small extent. This, in turn, to constitute a comprehensive study of the development of a new product should be significantly extended.

Kind regards 

Reviewer

Author Response

1 Introduction: I propose to broaden the thematic scope of this chapter. Important from the point of view of developing new products with the addition of cricket powder is a thorough and comprehensive market research. And information on the subject should be included in the introduction. In addition, it is also worth mentioning the obstacles to introducing cricket powder into food production. Is it regulated by law? What are the concerns? e.g. microbiological safety of such a raw material? Any antinutritional components?

Response: The regulated by law and the concerns have been added in the introduction part. “Novel foods must be authorized by the Commission Implementing Regulation (CIRs). Food Business Operators (FBOs) have been authorized to place defined types of Insects Based-Products (IBPs) on EU market. Of course, crickets were one of those insects [1].” And “Raw materials must come from hygienic farm which certified by the Department of Livestock Development.”

2 Material and Methods:

I would like more information about where the crickets came from (what farm, "age" of the crickets).

Response: The source of cricket has been added in 2.1. “Crickets were provided from Good Agricultural Practice (GAP) certified farm in Khon Kaen province, Thailand, indicating hygiene and safety from dangers and con-tamination.”

3 Hardness measurements were taken. Were crispiness measurements taken of the chips?

Response: The measurement of crispiness showed a wide standard deviation due to the non-uniform composition of the product. Crispiness measurement involves assessing fracture behavior, the number of fracture points, and the sound produced during breakage, thus multiple factors influence quality analysis. For sample comparison in this experiment, the maximum force required to cause fracture (hardness) was used to explain the differences between samples containing varying amounts of cricket powder in the chips.

4 How was the product described for sensory testing? Were respondents recruited using a recruitment form to identify a group of respondents with a positive/moderate attitude towards the introduction of cricket powder into food products?

Response: The sensory panelists had not been previously trained; however, for the sensory evaluation, explanatory documents detailing the procedure and test samples were provided. All panelists voluntarily participated in the sensory tests.

5 Storage tests (6 months) should also concern sensory characteristics, water content, water activity, and colour and texture.

Response: In the storage experiment, moisture content, total microbial counts, and color of the products were tested. The results showed that moisture content did not differ significantly among samples, and microbial counts remained within acceptable standards. This information will also be added to the Materials and Methods and Results and Discussion sections.

6 Results and Discussion: The discussion of the results is correct, however, following the comments on the methods, it is necessary to broaden the issue of storage studies. Such research and its results would contribute to proposing an optimal method of packaging this product. The authors focus only on TBARS and colour measurements.

Response: In the storage test of this chips product, moisture content and total microbial counts were examined. It was found that the moisture content was very low (1%), with no significant difference from the product before storage, which corresponded with the absence of microbial growth. In the storage of snack products, a major factor affecting deterioration is rancidity. Since the experimental product contains low fat, confirmation is needed to verify that the rate of oxidation reactions is low. Meanwhile, color parameters are important for selecting appropriate packaging for storage; if there is significant color change, opaque packaging should be chosen to minimize browning reactions. This information will be added to the Results and Discussion section.

7 Conclusions: The conclusions in the text are a repetition of the discussion of the obtained results. In the meantime, at this point it should be concluded whether the used additive is promising as a component of the recipe for chips? To what extent does it change the nutritional value of the product at the level of the best consumer acceptability? What should be improved so that the share in the recipe is greater and at the same time the product gains consumer acceptance. And very important storage studies, to which the authors referred only to a small extent. This, in turn, to constitute a comprehensive study of the development of a new product should be significantly extended.

Response: Conclusions have been rephrased to “From the chemical composition analysis of cricket powder, it was found that freeze-drying resulted in the highest protein content. However, the antioxidant activity, analyzed through DPPH, showed no significant difference across the four drying methods (p>0.05). For commercial applications, freeze-drying is recommended for preparing cricket powder. However, for small-scale industrial processing, other drying methods may be used to help reduce costs. When cricket powder was applied in chips, it was found that increasing the amount of cricket powder decreased lightness and yellowness while increasing redness and hardness. The antioxidant activity and phenolic content of the chips in-creased with the amount of cricket powder added, as crickets have high antioxidant properties. Meanwhile, the flavonoid and potassium content decreased when vegetable powder was replaced with cricket powder. However, the chips containing a 10:10 ratio of vegetable powder to cricket powder received the highest satisfaction ratings in all attributes. This product can be stored safely for 6 months with safe oxidation reaction and no microbial growth. The formula that achieved the highest consumer satisfaction had high protein and fiber content, but a lower fat content. It also exhibited high antioxidant activity, making vegetable chips with cricket powder a promising healthy snack alternative and a functional food that meets the demands of future food trends.”

Reviewer 3 Report

Comments and Suggestions for Authors

The paper looks interesting by means of the aim of the study. Nowadays the economy as well as food attitude is changing so alternative protein sources are needed and needs a detailed research. Authors proposal here is chips made from kale with insect flour addition. However the work is well planned the range of experiments is basic. It allows to show some trends but the paper suffer from lack of discussion. The discussion is mainly done by means of comparison with other sources rather than explaining phenomena observed. According to that I designate the paper as needed major revision. To help Authors in improving the text the list of main hot spots that’s needs to be fixed is listed below.

  1. What was the temperature of the material (crickets) during microwave drying?
  2. If crickets flour are added to the chips they are no more “vegetable chips” it should be corrected
  3. What is Na(NO2)3 (L161) and AlNO3 (L162). Please correct formulas
  4. Sensory evaluation is a research done on humans. Did authors possess the any Institutional Review Board Statement. It was not indicated in the manuscript
  5. The differences in chemical composition of cricket flour obtained using 3 methods of drying (Table 2) should be explained, not only compare to references
  6. Figure 1 should be strongly improved
  7. Figure 4 should be strongly improved (add measurement points)

Author Response

1 What was the temperature of the material (crickets) during microwave drying?

Response: Crickets contain 60–70% moisture. When drying 100 g of crickets in each drying cycle, the internal temperature ranged between 80–90 °C.

2 If crickets flour are added to the chips they are no more “vegetable chips” it should be corrected

What is Na(NO2)3 (L161) and AlNO3 (L162). Please correct formulas

Response: The vegetable chips have been revised to chips throughout manuscript. The formulas have been corrected.

3 Sensory evaluation is a research done on humans. Did authors possess the any Institutional Review Board Statement. It was not indicated in the manuscript

Response: The ethics information has been added in sensory evaluation part. “This research has been approved by human research ethics from the Khon Kaen University Ethics Committee for Human Research (KKUEC) in KKUEC’s Exemption Determination Regulations No. HE68018.”

4 The differences in chemical composition of cricket flour obtained using 3 methods of drying (Table 2) should be explained, not only compare to references

Figure 1 should be strongly improved

Figure 4 should be strongly improved (add measurement points)

Response: The chemical composition has been explained. “The fat content of the cricket powder ranged from 26.85-28.98%, with the micro-wave drying method resulting in the highest fat content because more water evaporated than other methods due to higher temperatures, while other methods did not show significant statistical differences (p>0.05) which has an inverse relationship with moisture content.” “Ash content indicated the mineral content in the cricket powder, and some minerals could be destroyed by heat. Thus, freeze-drying resulted in the highest ash content, followed by vacuum drying, hot air drying, and microwave drying, respectively.”

The Figure 1 and 4 have been revised to table.

Reviewer 4 Report

Comments and Suggestions for Authors

This manuscript addresses an important topic—using cricket powder as a sustainable protein source in vegetable snacks—and employs a comprehensive experimental design. Overall, the article has certain novelty and advantages for this field of research work. However, some issues still need to be improved.

Keywords: The current keywords are too generic. Please add more distinctive terms to enhance discoverability.

Abstract coherence: Although the abstract covers the main findings, it reads like a list of results and lacks cohesive summary language. Please restructure the abstract to create a clear, logical narrative.

Abstract background: The abstract omits any statement of research background or significance and jumps directly into experimental results. Please introduce a brief opening sentence that explains the study’s context and objectives.

Introduction logic: The introduction cites many relevant studies but fails to form an organic logical thread, making the justification for this work appear weak. Please reorganize the introduction so that each paragraph naturally leads to the next and clearly conveys the necessity and novelty of this research.

Objective framing: Although the research aim is clearly stated, it remains descriptive. Please reformulate the objectives as hypothesis‑driven, question‑oriented statements

Methods description: In the “Preparation of cricket powder” section, the use of numbered steps (1.–4.) improves logical order but hinders readability. Please revise this section—either by integrating steps into concise prose or summarizing them in a table—to improve clarity.

Replication details: The “Statistical analysis” section should specify the number of replicates for each measurement (e.g., color readings, hardness tests) to ensure reproducibility.

Figure 1 integration: In Section 3.2, the link between Figure 1 and the discussion is not sufficiently explicit. Please strengthen the in‑text references to Figure 1 and provide a clear interpretation of its key trends.

Color data presentation: In Section 3.4, while bar charts effectively display L*, a*, and b* values, comparing single variables across treatments may be more transparently presented in a table. The same suggestion applies to Figure 5.

Conclusions scope: The conclusion section merely repeats the results and lacks discussion of broader implications. Please expand this section to address practical significance and propose specific future research directions.

 Reference formatting: Please verify that all DOIs are correct and that journal names and abbreviations conform to the target journal’s style guidelines.

Author Response

1 Keywords: The current keywords are too generic. Please add more distinctive terms to enhance discoverability.

Response: Protein source and antioxidant have been added in keywords.

2 Abstract coherence: Although the abstract covers the main findings, it reads like a list of results and lacks cohesive summary language. Please restructure the abstract to create a clear, logical narrative.

Response: Abstract has been rephrased.

3 Abstract background: The abstract omits any statement of research background or significance and jumps directly into experimental results. Please introduce a brief opening sentence that explains the study’s context and objectives.

Response: Background and objectives have been added in abstract. “This research was based on the low acceptance of edible insects, so the development of protein supplements from crickets was interesting. The objectives of this research were to study the effects of different drying 4 methods on the chemical properties and were to study the effects of cricket powder fortification in vegetable chips on the chemical and physical qualities and consumer acceptance.”

4 Introduction logic: The introduction cites many relevant studies but fails to form an organic logical thread, making the justification for this work appear weak. Please reorganize the introduction so that each paragraph naturally leads to the next and clearly conveys the necessity and novelty of this research.

Response: The regulated by law and the concerns have been added in the introduction part. “Novel foods must be authorized by the Commission Implementing Regulation (CIRs). Food Business Operators (FBOs) have been authorized to place defined types of Insects Based-Products (IBPs) on EU market. Of course, crickets were one of those insects [1].” And “Raw materials must come from hygienic farm which certified by the Department of Livestock Development.”

5 Objective framing: Although the research aim is clearly stated, it remains descriptive. Please reformulate the objectives as hypothesis‑driven, question‑oriented statements

Response: The objectives have been revised. “This research aims to explore the effects of different drying 4 methods that were hot air drying, vacuum drying, microwave drying, and freeze-drying on the chemical qualities and to study effects of cricket powder fortification in vegetable chips on the chemical and physical properties and consumer acceptance.”

6 Methods description: In the “Preparation of cricket powder” section, the use of numbered steps (1.–4.) improves logical order but hinders readability. Please revise this section—either by integrating steps into concise prose or summarizing them in a table—to improve clarity.

Response: The preparation of cricket powder has been rephrased. “Crickets were provided from Good Agricultural Practice (GAP) certified farm in Khon Kaen province, Thailand, indicating hygiene and safety from dangers and contamination. The crickets processed into adult crickets were 40-45 days old. The crickets were first washed and evenly distributed on trays, followed by drying using one of the following methods: (1) hot air oven drying at 60 °C for 48 h, (2) vacuum drying at 55 °C for 6 h, (3) microwave drying at 550 W for 6.5 min, and (4) freeze-drying. After drying, the crickets were finely ground using a mechanical grinder and passed through a 200 mesh sieve to obtain cricket powder. The resulting powder was stored at 4 °C until further use in subsequent experiments.”

7 Replication details: The “Statistical analysis” section should specify the number of replicates for each measurement (e.g., color readings, hardness tests) to ensure reproducibility.

Response: The replication details were added in statistical analysis part. “The color analysis and hardness have been measured 5 and 10 repetitions, respectively. Whereas, the chemical analysis and preservation testing have been measured 3 repetitions per sample.”

8 Figure 1 integration: In Section 3.2, the link between Figure 1 and the discussion is not sufficiently explicit. Please strengthen the in‑text references to Figure 1 and provide a clear interpretation of its key trends.

Response: Figure 1 has been changed to Table 4. Section 3.2 has been link with Table 4 and rephrased. “A sensory evaluation of chips by 30 panelists found that the overall preference of all samples were not significant difference (p>0.05) but the chips without cricket powder were most accepted in terms of appearance and color, as they had the brightest green color. However, there was no significant difference from the sample with 15:5 and 10:10 ratio of vegetable powder to cricket powder (Table 4). On the other hand, the chip with a kale:cricket powder ratio of 0:20 received the lowest satisfaction scores for appearance and color, with statistical significance (p<0.05), as the cricket powder resulted in a brown color. This result was consistent with the color measurement findings, where lightness (L*) and yellowness (b*) values decreased while redness (a*) increased as the amount of cricket powder increased. Cricket powder chips received the ranging from neutral to slightly liked in all sensory attributes. Meanwhile, the chips containing a 10:10 ratio of vegetable powder to cricket powder, or 20% each of vegetable and cricket powder of the total dry ingredients, were the most liked in all sensory attributes, ranging from liked to very liked. Specifically, the flavor of the kale:cricket chip with a 10:10 ratio received significantly higher scores (p<0.05) compared to the samples with 20:0 and 15:5 ratios. As for texture satisfaction, it varied initially increasing and then decreasing with higher amounts of cricket powder, with the 10:10 ratio chip achieving the highest score.”

9 Color data presentation: In Section 3.4, while bar charts effectively display L*, a*, and b* values, comparing single variables across treatments may be more transparently presented in a table. The same suggestion applies to Figure 5.

Response: Figure 2 and 5 have been changed to Table 7 and 8.

10 Conclusions scope: The conclusion section merely repeats the results and lacks discussion of broader implications. Please expand this section to address practical significance and propose specific future research directions.

Response: Conclusions have been rephrased. “From the chemical composition analysis of cricket powder, it was found that freeze-drying resulted in the highest protein content. However, the antioxidant activity, analyzed through DPPH, showed no significant difference across the four drying methods (p>0.05). For commercial applications, freeze-drying is recommended for preparing cricket powder. However, for small-scale industrial processing, other drying methods may be used to help reduce costs.

When cricket powder was applied in chips, it was found that increasing the amount of cricket powder decreased lightness and yellowness while increasing redness and hardness. The antioxidant activity and phenolic content of the chips increased with the amount of cricket powder added, as crickets have high antioxidant properties. Meanwhile, the flavonoid and potassium content decreased when vegetable powder was replaced with cricket powder. However, the chips containing a 10:10 ratio of vegetable powder to cricket powder received the highest satisfaction ratings in all attributes. This product can be stored safely for 6 months with safe oxidation reaction and no microbial growth.

The formula that achieved the highest consumer satisfaction had high protein and fiber content, but a lower fat content. It also exhibited high antioxidant activity, making vegetable chips with cricket powder a promising healthy snack alternative and a functional food that meets the demands of future food trends.”

11 Reference formatting: Please verify that all DOIs are correct and that journal names and abbreviations conform to the target journal’s style guidelines.

Response: Reference formatting has been corrected throughout the manuscript.

Round 2

Reviewer 1 Report

Comments and Suggestions for Authors

The Authors have sufficiently addressed the comments and concerns. No further revisions are necessary.

Author Response

Thank you for your suggestions and comments.

Reviewer 2 Report

Comments and Suggestions for Authors

Dear Authors,
Thank you for preparing responses to my comments on the manuscript. I have noticed that most of them have been taken into account, and the authors have added a comment to the remaining ones with justification regarding the scope of the experiment conducted. In my opinion, the weakest part of the entire text is the part concerning sensory and storage studies. Moreover, storage studies after selecting the appropriate scope of studies could constitute the second part of the implementation of this topic. Perhaps the authors are planning extensive storage studies in the future.

I do not make any further comments to the presented revised manuscript, but since it describes a food product, I recommend attaching a photograph of the product to the manuscript (minor revision).

Kind regards

Reviewer

Author Response

The photographs of products have been added in Figure 1 according to the reviewer's suggestion.